# Oral Mucosal Lesions in Childhood

**DOI:** 10.3390/dj10110214

**Published:** 2022-11-09

**Authors:** Lorena Horvat Aleksijević, Jelena Prpić, Miranda Muhvić Urek, Sonja Pezelj-Ribarić, Nataša Ivančić-Jokić, Romana Peršić Bukmir, Marko Aleksijević, Irena Glažar

**Affiliations:** 1Faculty of Dental Medicine and Health, Josip Juraj Strossmayer University of Osijek, 31000 Osijek, Croatia; 2Clinic of Dental Medicine, Clinical Hospital Center Rijeka, Krešimirova 42, 51000 Rijeka, Croatia; 3Department of Oral Medicine, Faculty of Dental Medicine, University of Rijeka, Krešimirova 40, 51000 Rijeka, Croatia; 4Department of Pediatric Dentistry, Faculty of Dental Medicine, University of Rijeka, Krešimirova 40, 51000 Rijeka, Croatia; 5Department of Restorative Dentistry and Endodontics, Faculty of Dental Medicine, University of Rijeka, Krešimirova 40, 51000 Rijeka, Croatia

**Keywords:** pediatric oral lesions, oral mucosa, children, hemangioma, lymphangioma, reccurent aphtous stomatitis

## Abstract

Childhood diseases are a continuous source of interest in all areas of general and dental medicine. Congenital, developmental, and hereditary diseases may either be present upon birth or appear in early childhood. Developmental anomalies, although often asymptomatic, may become grounds for different infections. Furthermore, they can indicate certain systemic disorders. Childhood age frequently brings about benign tumors and different types of traumatic lesions to the oral mucosa. Traumatic lesions can be caused by chemical, mechanical, or thermal injury. Mucocele and ranula are, by definition, traumatic injuries of the salivary glands or their ducts. Recurrent aphthous lesions are the most common type of ulcerations in childhood, and their etiology is considered multifactorial. Oral mucosal lesions in children require different treatment approaches depending on etiological factors and clinical presentation. Clinicians should have adequate knowledge of oral anatomy in order to diagnose and treat pathological conditions.

## 1. Introduction

A child’s oral mucosa should be pink, smooth, moist, and shiny in its appearance. The dorsum of the tongue should have a velvety surface texture, and the lingual frenulum should be of adequate length to allow normal tongue movements. Oral mucosal lesions in children may appear as changes in color, size, or structure of the normal oral anatomy [1]. Changes in a child’s mouth, besides causing parents’ concern, can lead to pain and discomfort; however, they can also be completely asymptomatic [2]. Typical anatomical structures such as linea alba or leukoedema may also be a cause of panic for parents at the time they first see them in their child’s mouth if they know nothing about their benign and asymptomatic character. This is where the knowledge and skills of the doctor of dental medicine become extremely important since they should be the ones to calm parents and explain the nature of these anatomical structures [1,3]. Congenital anomalies, such as ankyloglossia, may present a heavy challenge for the parents depending on its level since it can impede normal feeding from an early age or later on when it poses a great difficulty for the child due to the restrictions in normal speech [1,4]. Developmental anomalies should be detected as early as possible since, although asymptomatic, they may predispose the child to develop different complications such as infections or indicate the presence of certain syndromes such as Down syndrome, which is often coupled with a fissurated tongue [2,5]. Many hereditary diseases show first symptoms in the oral cavity. Recognition of these oral symptoms can lead to an early diagnosis and therapy before the disease spreads to other organ systems [1,6]. Benign tumors occur relatively often in the oral cavities of newborns and children. The most common is fibroma, a benign tumor of the connective tissue, followed by hemangioma and lymphangioma [1,2]. Traumatic lesions such as mucocele and ranula can be the underlying cause of swelling which will require surgical intervention [2,7]. Traumatic lesions can be classified according to the type of injury as thermal, mechanical, or chemical. Their treatment should be focused on enhancing the healing process in order to avoid complications such as infections, which would further aggravate the condition [1,8]. Recurrent aphthous lesions, which appear quite frequently in children, may impede the normal feeding process, swallowing, and speech [9]. The purpose of this review is to describe common soft tissue lesions of the oral cavity in children in order to help timely diagnosis and treatment; furthermore, we will also address some less frequent changes which may indicate the presence of systemic disorders which require a multidisciplinary approach.

## 2. Physiological Structures

Physiological structures are very common, benign and asymptomatic lesions of oral mucosa and do not require treatment but should not be mistakenly diagnosed as pathological lesions [10].

### 2.1. Linea Alba

This benign condition is typically localized on the buccal mucosa and stretches from the labial commissure towards the molar region [10]. It is clinically easily recognized as a white line of different intensity and thickness located at the level of occlusal surfaces [1,3,10]. It can be found either unilaterally or bilaterally [3]. Occasionally it may appear on the lateral borders of the tongue. It does not require treatment [1,10].

### 2.2. Leukoedema

Leukoedema is a common, benign, and asymptomatic lesion of the oral mucosa, which is considered a variation of the normal mucosal anatomy [3,10,11]. It appears as a whitish lesion located bilaterally or unilaterally on the buccal or labial mucosa [1,11]. It is of unknown etiology and is more frequently found in African Americans and among males [1,3,10]. Clinical presentation is of diffuse white creases or patches which disappear once the mucosa is stretched [10]. Since this lesion is benign and asymptomatic, it does not require treatment [1,10].

## 3. Congenital Anomalies

Congenital anomalies are usually benign anomalies which can affect a child’s function or aesthetics. They should be monitored because of their impact on the quality of a child’s life and properly diagnosed because of the possible malignant potential of nevi [1,2].

### 3.1. Ankyloglossia

Ankyloglossia is a congenital anomaly characterized by an abnormally short lingual frenulum which significantly limits tongue mobility [1,4,10,12,13,14]. A heart-shaped invagination at the tongue tip forms when tongue elevation is attempted [13,15]. In newborns, a short frenulum can follow the child’s growth and cease to represent a functional difficulty when the child reaches a certain age [2]. Variability in the position and insertion level of the lingual frenulum has been observed; therefore, in order to reach the correct diagnosis, the determination of the functional disorder is of prime importance rather than anatomical variability [4,16]. If the child can lick his/her lower lip, it is considered that there are no functional disturbances and treatment is not indicated [2,12,16]. More severe types of ankyloglossia result in breastfeeding problems and impaired speech development later on [4,10,14,15,16]. In those cases, surgical therapy, frenectomy, is indicated, whereas, in cases of impaired phonatory function, treatment must also include the help of a logopedist [2,10,13,14].

### 3.2. Congenital Epulis

Congenital epulis is also known as the granular cell tumor [17,18,19]. It is a rare and benign lesion found in newborns and may often be found even before birth [17,18,20,21]. It is more common in girls [18,21], with typical localization on the alveolar ridge of the upper jaw, although it can also be found in the mandible [17,20]. Clinically it presents as a pedunculated nodule of the same color as the surrounding mucosa, elastic and smooth surfaced [19,21]. It usually measures approximately 10 mm in diameter, and it has been demonstrated that it does not follow the child’s growth, but rather it remains of the same size; therefore, at follow-up, it appears smaller [21]. The treatment of choice is usually surgical, and diagnosis is confirmed through pathohistological analysis [18], which shows the proliferation of large eosinophilic polygonal cells with eccentric nuclei and granular cytoplasm [17,20,21].

### 3.3. Melanocytic Nevus

Melanocytic nevus is a pigmented mucosal lesion which is caused by the accumulation of pigment-producing cells called melanocytes [4,15,22]. It can be congenital or develop at any time during life [4,15,22,23]. Histological classification of nevi is crucial since it determines their prognosis (Table 1) [1]:Junctional nevus: the proliferation of melanocytes in proximity to the network of blood vessels and nerves located superficially. It is usually limited to the epithelium [1,24].Compound nevus: the proliferation of melanocytes in both epithelium and the underlying connective tissue [24].Intradermal/intramucosal: melanin-producing cells that are located in lamina propria and are not in contact with the basal membrane. The lesions are typically dome-shaped, light brown, and located on the gums, lips, or buccal mucosa [1,24].Blue nevus: the proliferation of elongated melanocytes deep within the lamina propria, far from the epithelium. This lesion is typically found on the hard palate [1,24,25]. They can be further classified into atypical blue nevus, locally aggressive blue nevus and congenital giant melanocytic nevus with nodular growth [1,25].Other melanocytic nevi include combined nevus and Spitz nevus, with palate or tongue localization [1,26].Congenital melanotic nevus: can be junctional, compound, intradermal, or intramucosal. They have their onset at birth, and they differ from common acquired nevi by their size and depth of involvement by nevus cells and adnexal and vascular involvement [1,27].

**Table 1 dentistry-10-00214-t001:** Prognosis and treatment according to histological type of nevi.

Type of Nevi	Prognosis and Treatment	Reference
Junctional nevus	Good prognosis. No treatment needed. May be surgical, cryotherapy, or laser therapy.	[28]
Compound nevus	Good prognosis. Surgical excision is treatment.	[29]
Intramucosal nevus	Good prognosis. Surgical excision is treatment.	[30]
Blue nevus	Possible malignancy. Pathohistological diagnosis is necessary.	[31]
Spitz nevus	Good prognosis in children.	[32]
	The diagnosis of a Spitz nevus must be carefully distinguished from melanoma. Surgical excision is treatment.	

The prevalence of oral nevi in children is unknown; however, solitary nevi are considered a relatively rare occurrence [1]. The most frequently observed nevi in the oral cavity are intramucosal and blue nevus, while compound nevus is the least common [33]. Melanocytic nevi present as localized brown, blue, grey, or black macules or papules of 0.1 to 3.0 cm in diameter [1,34]. They are asymptomatic and are usually found by accident upon clinical examination [33]. Nevi are commonly localized on the hard palate, buccal mucosa, and gums [1,34]. They are extremely rare in the retromolar area [33]. Diagnostic procedures include excisional biopsy with the exception of mucosal melanoma, especially if the lesion is localized on the palate [1,33,34]. At the same time, excisional biopsy also poses as the treatment [1,33].

## 4. Developmental Anomalies

Developmental anomalies are relatively common in children [1,11]. Although their cause is unknown, they can be caused by hereditary factors or occur as a symptom in various syndromes [2,5,10].

### 4.1. Geographic Tongue

Benign migratory glossitis, often called geographic tongue (Figure 1), appears in 1–3% of the population and is not uncommon in children [10,35,36]. The cause is unknown; however, it is assumed that a significant role is played by hereditary factors [1,10,36,37]. The disorder is also often related to various systemic and psychological conditions [36,38]. Geographic tongue is marked by erythematose, round or irregularly shaped patches on the dorsal and lateral portions of the tongue [2,11,36,38]. The sides of the tongue are slightly elevated, hyperkeratotic, and yellowish [2,10,36,37]. Lesions change their position on the tongue surface over time; therefore, they have been termed “migratory“ [2,35,36]. Areas of desquamation are prone to secondary infections, so the inflamed areas may become quite painful, although this is rarely the case; typically, this condition is asymptomatic and resolves spontaneously [2,10,35]. In cases of pain or discomfort, the application of antiseptics, topical anesthetics, and/or corticosteroids can be indicated [10,35].

### 4.2. Fissured Tongue

Fissured tongue is a developmental anomaly which is typically presented as a solitary anteroposterior fissure (groove) right in the middle of the dorsal surface of the tongue [11]. It is not uncommon to find smaller and shallower furrows originating from the main fissure and spreading radially [2,10,11,35]. Although it represents a more frequent finding in adults, fissured tongue may appear in children in the form of an isolated developmental anomaly or coupled with other disorders such as Down syndrome and Melkersson–Rosenthal syndrome [2,5,10,11,35]. Melkersson–Rosenthal syndrome, although extremely rare in everyday clinical settings, is characterized by a triad of symptoms-orofacial edema, hemifacial paralysis, and fissured tongue [5,39]. Fissured tongue is often related to complications in the form of inflammation and secondary fungal infections as a consequence of food debris retention in deeper grooves [2,10,11,35,39].

### 4.3. Retrocuspid Papilla

Retrocuspid papilla is one of the developmental anomalies that may be found in many children [11]. It is located on the attached gingiva on the lingual aspect of lower canines and typically occurs bilaterally [2,11]. This solid, fibro-epithelial pink to red papula [40], measuring 2 to 3 mm in diameter, is usually asymptomatic and has a tendency to decrease over time; therefore, it does not require treatment [2,40].

## 5. Hereditary Diseases

Hereditary diseases such as white sponge nevus, Peutz–Jeghers syndrome, or neurofibromatosis type 1 may present with oral symptoms that are important for establishing a final diagnosis and, accordingly, appropriate treatment [1,11,41,42,43].

### 5.1. White Sponge Nevus

White sponge nevus is a benign asymptomatic lesion which is inherited as an autosomal dominant disorder [1,11,41,44,45,46,47,48,49]. Lesions are clinically presented as uni- or bilateral white patches of thick, sponge-like, or velvety tissue which are non-scrapable [1,11,44,46]. They are most commonly found on the buccal mucosa but may also be located on the tongue surface, labial mucosa, mouth floor, and gingiva [1,46,49,50]. Usually, they are already present at birth or in early childhood and occasionally may develop during adolescence [1,41,45,47,50,51]. Differential diagnosis includes leukoplakia, chemical burns, trauma, irritation caused by tobacco smoke, and candidosis [1,11,47,48,51]. Treatment is not required unless mastication is compromised [1,41,49].

### 5.2. Peutz–Jeghers Syndrome

Peutz–Jeghers syndrome is an autosomal dominant disorder characterized by gastrointestinal polyposis and dark-colored spots on the skin and mucosa [1,11,42,52,53,54,55]. Hamartomatous polyps in the gastrointestinal tract can cause abdominal pain, chronic bleeding, anemia, and obstruction of the intestines [6,42,52,53,54,55], whereas 2 to 3% of the polyps show a tendency towards malignant transformation [1,6]. Polyps in the gastrointestinal tract may develop at any age, but pigmentations usually occur in early childhood [6,54,56]. Skin lesions are most commonly found around the eyes, on the fingers, and around the mouth, while intraorally, they are typically localized on the buccal mucosa and inner side of the lips [1,6,52,54,55,56]. Lesions are round or oval, 2–5 mm in diameter [1,6,25]. Their color varies from dark brown to black [6,25,56]. Lesions are asymptomatic, and the majority of intraoral ones fade before the first decade of life [52,54]. However, it is of major importance to diagnose the described changes in a timely manner and refer the patient to a gastroenterologist due to the possible progression of hamartomas towards malignancy [1,11,25,42,54,55].

### 5.3. Neurofibromatosis Type 1

Neurofibromatosis type 1, also referred to as von Recklinghausen’s disease, is an inherited autosomal dominant disorder characterized by the growth of multiple benign tumors along the nerves and on the skin, neurofibromas [11,43,57,58,59]. In cases when it is localized on the head and neck, it usually affects the skin; however, neurofibromas in the mouth are not uncommon [43,57,59,60,61,62]. There, it is typically present in the form of a submucosal, soft, discreet mass of smaller diameter, mostly on the alveolar processus and palate [11,43,57,59,60,61,62,63,64]. Neurofibromatosis type 1 should be suspected in cases when the described changes are associated with multiple café-au-lait spots on the skin [11,43,57,58,59,62,64].

## 6. Benign Tumors

Benign tumors are relatively often in oral cavities of newborns and children. The most frequent is fibroma, a benign connective tissue tumor, followed by hemangioma and lymphangioma [1,2].

### 6.1. Fibroma

One of the most common benign lesions of the oral cavity, fibroma, results from connective tissue proliferation, which is brought about by chronic irritation [1,2]. Fibromas can appear anywhere on the oral mucosa; however, they are typically located on the palate, tongue, buccal mucosa, or lips [2,65,66,67]. Most fibromas are less than 1 cm in diameter, their color does not differ much from the surrounding mucosa, and the tissue feels smooth and hard [1,2,65]. They can be pedunculated or dome-shaped and firmly attached to the base [2,65]. Treatment includes surgical removal and elimination of the source of irritation. Relapses are rare [2].

### 6.2. Hemangioma

Hemangiomas are benign, fast-growing, vascular hamartomas which may appear anywhere on the soft tissues but most frequently develop on the buccal mucosa, dorsum of the tongue, gums, and lips [2,66,68,69,70,71,72,73]. They are relatively frequent in children [1,2,66,71,72,74]. As for their clinical presentation, they have typical red color if they are localized closer to the surface; however, those located more deeply appear blue [1,2,73,75]. They protrude upward from the surface and feel moderately hard upon palpation. Hemangiomas appear very early in life and are more common in girls [2,70,72,74]. As for growth, they develop fast and progressively follow the child’s growth dynamic [71]. They are usually painless but can ulcerate or bleed due to trauma. Treatment is by laser or surgical resection [2,68,71,72,73,74]. Due to the vascular nature of the tumor, the danger of excessive bleeding must be taken into consideration [2,68,75].

### 6.3. Lymphangioma

Lymphangiomas are benign tumors of the lymphatic system, usually present at birth, although they may also develop during infancy [2,72,76,77]. Intraorally they are most commonly found on the tongue but may also be located on the lips and buccal mucosa [2,72,77]. Superficially located tumors are soft upon palpation, pink, or red/bluish, while more deeply localized lesions may remain invisible from the surface [2,72]. Cystic hygroma is a sac-like large lymphangioma which may involve the tissues of the mouth floor and neck [2]. Every lymphangioma that presents a functional or esthetic defect should be removed surgically [2,72,77]. Relapses are quite frequent and are usually caused by the lack of a cyst wall [2].

## 7. Traumatic Lesions

Mucoceles and ranulas are one of the most common salivary gland disorders, and they are classified as extravasation pseudocysts [78].

### 7.1. Mucocele

Mucocele develops as a consequence of mechanical trauma to a minor salivary gland [2,7,65,78,79,80,81,82,83], which is followed by saliva retention and accumulation inside the blocked and dilated excretory ducts of the gland [2,7,65,78,79,80,81,83,84,85]. Lesions are usually painless, with smooth surfaces, bluish or transparent [2,7,65,72,78,79,80,82,83,85]. Most are not larger than 1 cm in diameter. They are treated by surgical removal; at that time, the surgeon often decides to perform the ablation of the neighboring minor salivary glands in order to prevent relapses [2,7,72,80,81,83].

### 7.2. Ranula

Ranula shows many clinical similarities with mucocele. It is caused by trauma to the excretory duct of the salivary glands located in the floor of the mouth and is manifested as swelling [7,72,78,84,86]. It is very uncommon in newborns [7].

## 8. Injury to the Oral Mucosa

An injury to the oral mucosa can result from physical, chemical, or thermal trauma. Thermal injuries are typically located on the anterior edge of the palate due to the consumption of hot foods [2]. Mechanic damage is caused by conscious or unconscious self-induced trauma. The most common example of self-induced physical trauma is morsicatio [10]. Morsicatio is a condition caused by chewing or rubbing of the mucosa brought about by stress or psychological disorders. Lesions are most frequently found on the lips, buccal mucosa, and tongue [8,10]. Clinically they present as white to grayish patches with a smooth or rough surface and irregular borders [1,8]. Treatment is not required, save for the elimination of the compulsive habit [1,10].

## 9. Recurrent Aphthous Lesions

Recurrent aphthous lesions (or ulcerations) are the most frequent kind of ulcerations observed in children [15,87,88]. They appear on the oral mucosa in the form of smaller or larger, single or multiple painful ulcerations that recur in intervals [9,89]. Many factors are involved in the etiology, including immune system disorders, genetic factors, hormonal disbalance, chemical, microbial or physical irritation, allergic factors, and stress [9,87,88,89,90,91,92,93,94]. Clinical presentation includes the formation of round or ovoid lesions with well-defined margins, a necrotic center covered by a yellow–gray pseudomembrane, and an erythematous halo which is the sign of peripheral inflammation [9,87,88,89,92,95]. Considering the size, number, and duration of the lesion(s), three types of aphthous ulcerations may be differentiated:Minor aphthae: lesions are typically less than 1 cm in diameter; they heal without scarring within 10 days (Figure 2).Major aphthous ulcerations: more than 1 cm in diameter, they can last for up to 30 days and may leave scars.Herpetiform aphthous ulcerations: multiple lesions, up to 3 mm in diameter; ulcerations may merge. Healing takes approximately 15 days [15,87,88,90,91,92,93,96,97].

Differential diagnosis is aimed at differentiation between aphtae and herpetic gingivostomatitis, herpangina, and ulcerations caused by injury [9]. Treatment is usually symptomatic and implies the use of topical anesthetics for pain control, antiseptic mouthwashes to prevent secondary infections, and products that promote re-epithelialization [9,92,96]. Topical corticosteroids may also be applied; however, only in older children [9,88,89,91,92,93].

## 10. COVID-19 Infection

Given that COVID-19 is a relatively new infection, the prevalence of oral lesions in COVID-19 infection, especially in children, is not known. The most frequently recorded oral lesions are blisters, ulcerations, and desquamating gingivitis. Ulcerations usually affect the dorsum of the tongue [98]. The presence of white plaques on the tongue that did not respond to local therapy and geographic tongue was also noted. Fungal infections, *Herpes simplex*, and *Herpes zoster* virus infections occurred as a result of stress and decreased immunity during the COVID-19 infection [98,99]. The Kawasaki-like symptoms, which include erythema, dryness, cracking, and bleeding of the oral mucosa, have been described as the most severe oral symptom [98].

## 11. Conclusions

Searching through the literature provides insight into the need for the thorough knowledge that doctors of dental medicine should have on the anatomy and pathology of the oral tissues. They should be able to recognize benign, asymptomatic, and typical anatomical structures and differentiate them from pathological conditions. Congenital and developmental anomalies, although often benign and asymptomatic, may cause difficulties in normal function, may be combined with different complications, or point to certain systemic disorders. Hereditary diseases often show intraoral symptoms before manifesting in other organ systems, which makes early diagnosis even more important and rather crucial. Benign tumors of the oral cavity usually require simple surgical intervention, just like complications of traumatic lesions, while other lesions caused by injury disappear spontaneously upon removal of the cause.

## Figures and Tables

**Figure 1 dentistry-10-00214-f001:**
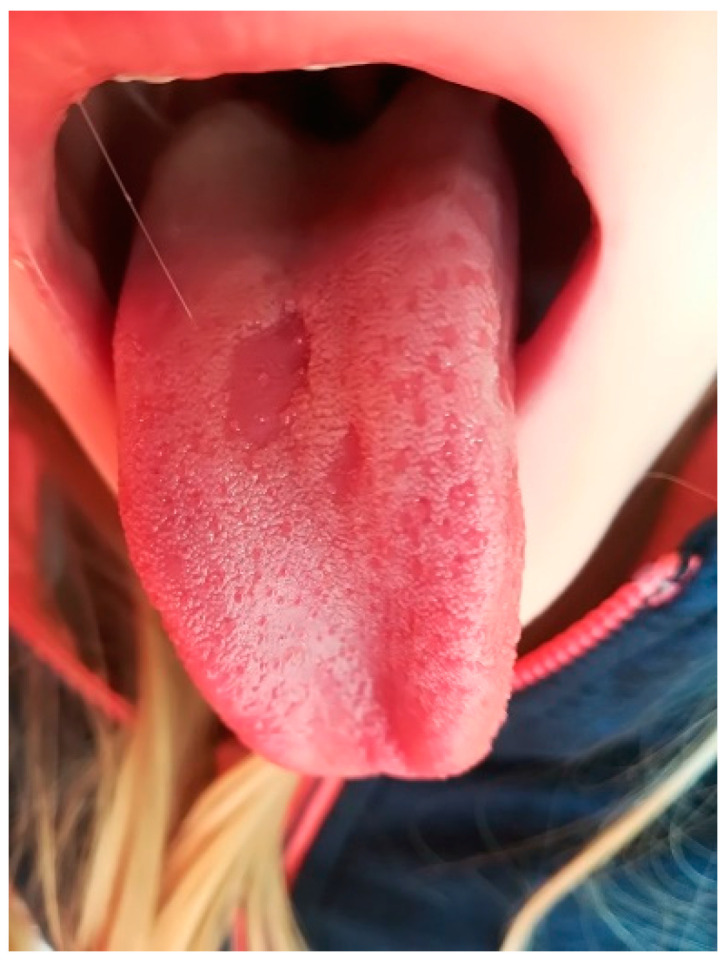
Geographic tongue.

**Figure 2 dentistry-10-00214-f002:**
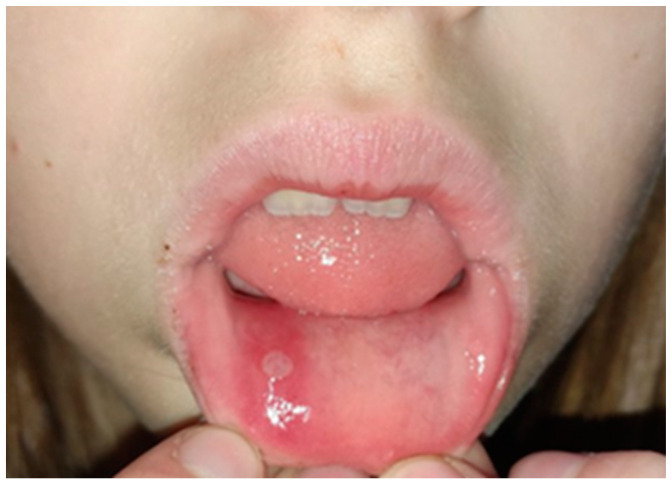
Aphtae minor.

## Data Availability

Not applicable.

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
