# Peer review of "Oral Mucosal Lesions in Childhood"

_dentistry, 2022, doi:10.3390/dj10110214_

Round 1

Reviewer 1 Report

Dear Authors,

thank you for giving me the opportunity to read and evaluate your work. I find it a great comprehensive review for the use i.e. by dental Students, and specialising doctors from oral mucosa diseases, and also pediatrics, since this could help with their work.
I find some minor changes to be possibly made:

1)  Could you find another name for the paragraph "2. Normal anatomical structures"? What is normal?

2) Please provide short introduction to all the chapters that are missing it (2,3,4,5 etc.) One-two sentences would suffice. 

3) Please provide the information wheter the pictures that have been taken have patients, and parents or guardians consent?

4) If you find the time could you provide the short subchapter about how the possible changes in the oral mucosa are affected by the COVID-19 disease in children? There has been recently one work that could help with the process if authors would want to go with this suggestion: DOI: 10.17219/dmp/131989

With best regards,

Reviewer

Author Response

Dear reviewer,

thank you for taking the time to read our manuscript, and also your comments and suggestions.

1) Could you find another name for the paragraph "2. Normal anatomical structures"? What is

normal?

We rephrased it to ''physiological''.

2) Please provide short introduction to all the chapters that are missing it (2,3,4,5 etc.) One-two

sentences would suffice.

We provided introduction to all the chapters that were missing it.

3) Please provide the information wheter the pictures that have been taken have patients, and parents or guardians consent?

We provided information.

4) If you find the time could you provide the short subchapter about how the possible changes in the

oral mucosa are affected by the COVID-19 disease in children? There has been recently one work that

could help with the process if authors would want to go with this suggestion: DOI:

10.17219/dmp/131989

We added a small chapter, since it is interesting topic, but we have manuscript related to infectious diseases of children (which will cover COVID-19 infection also) in preparation.

Best regards,

Authors

Reviewer 2 Report

I would like to congratulate the authors for their submission. This an interesting review summarizing the common oral mucosal diseases in children. Indeed, knowing the manifestation of oral mucosal diseases is important for caregivers and parents alike. It should be acknowledged, however, that syndromic/hereditary predisposed oral mucosal lesions are enormous than what is already mentioned in the review. Nevertheless, I do not have any comments regarding the scientific validity in this submission, however, it will greatly benefit from an English editing to correct typos and enhance sentence structure.

Author Response

Dear reviewer,

Thank you for taking the time to read our manuscript.

Thank you for your comments. An English language professor did extensive English language proofreading.

Best regards,

Authors